# Refined Dual-Phase-Lag Theory for the 1D Behavior of Skin Tissue under Ramp-Type Heating

**DOI:** 10.3390/ma16062421

**Published:** 2023-03-17

**Authors:** Ashraf M. Zenkour, Tareq Saeed, Amal M. Aati

**Affiliations:** 1Department of Mathematics, Faculty of Science, King Abdulaziz University, Jeddah 21589, Saudi Arabia; 2Department of Mathematics, Faculty of Science, Kafrelsheikh University, Kafrelsheikh 33516, Egypt; 3Financial Mathematics and Actuarial Science (FMAS)-Research Group, Department of Mathematics, Faculty of Science, King Abdulaziz University, Jeddah 21589, Saudi Arabia; 4Department of Mathematics, College of Science and Arts and Applied College Branch in Rijal Alma’a, King Khalid University, Abha 61411, Saudi Arabia

**Keywords:** refined DPL theory, bio-thermal, skin tissue, ramp-type heating, 74F05, 74J05, 92B05

## Abstract

In this article, a mathematical analysis of thermoelastic skin tissue is presented based on a refined dual-phase-lag (DPL) thermal conduction theory that considers accounting for the effect of multiple time derivatives. The thin skin tissue is regarded as having mechanically clamped surfaces that are one-dimensional. Additionally, the skin tissue undergoes ramp-type heating on its outer surface, whereas its inner surface keeps the assessed temperature from vanishing. Some of the previous generalized thermoelasticity theories were obtained from the proposed model. The distributions of temperature, displacement, dilatation, and stress are attained by applying the Laplace transform and its numerical reversal approaches. The outcomes are explicitly illustrated to examine the significant influences on the distributions of the field variables. The refined DPL bioheat conduction model in this study predicts temperature, and the findings revealed that the model is located among the existing generalized thermoelastic theories. These findings offer a more thorough understanding of how skin tissue behaves when exposed to a particular boundary condition temperature distribution.

## 1. Introduction

Recently, heat therapy has begun to take on increased significance and effectiveness as a biomedical treatment method. Experimenting is the best way to learn about the thermomechanical behavior of organic tissue. Nevertheless, due to the difficulty of physical and biochemical procedures and the range of tissue trials, such as the sample location of a solid and the sampling one, it is rather challenging to carry out an entire experiment. Thus, a general theoretical evaluation of the procedure is crucial and needed for the estimate of thermomechanical behaviors to increase the efficacy of thermal therapy. A small shift in stress and temperature in biological tissue will vary the hormone production rate, disable the protein, and even influence the immune system to some level [1]. Under different thermal loadings in contemporary medicine and daily life, such as ultrasound, laser, burns, etc., organic tissues demonstrate diverse thermomechanical behaviors prophesied by generalized thermoelasticity theories via the coupled impact between temperature and dilatation, which does not happen in classical non-coupled thermoelasticity.

The way heat moves through skin tissue is very complicated and involves many different mechanisms such as blood flow, tissue properties, heat conductivity, and metabolic heat generation. Understanding how heat affects skin tissue is important for medical treatments, which makes bioheat models very significant. Many different models explain how skin tissue responds to heat: Pennes [2], classical coupled thermoelasticity (CTE) [3], Lord–Shulman (L–S) [4], Green–Lindsay (G–L) [5], Tzou [6], and Chandrasekharaiah [7] models.

The responses of the human integumentary and musculoskeletal systems to heat sources that are used directly have been assessed in many studies. Xu et al. [8] conducted a comprehensive literature review of the biothermo-mechanical behavior of skin tissue. They examined four subject areas: (i) skin structure, (ii) bioheat transfer and thermal damage, (iii) biomechanics, and (iv) bio-thermomechanics. Singh and Melnik [9] reviewed the effectiveness of ablation techniques use of heat to remove or destroy diseased tissue in the body and discussed the two popular methods radiofrequency ablation (RFA) and microwave ablation (MWA). Andreozzi et al. [10] provided an overview of the changes that have been made to bioheat models over time and discussed their importance for predicting temperature distribution in tissues during biomedical applications, due to ethical concerns limiting experimentation in this field. Penne’s equation was resolved by Khiavi et al. [11] in terms of appropriate thermal/physiological properties for sixteen solid parts and a controlling system was added to the Pennes equation by applying the thermoregulatory mechanisms of the 65-node Tanabe (65MN) model. In the context of a non-stable thermal transport pattern, Li et al. [12] described the thermal and mechanical responses of porous organic tissue exposed to an instant thermal shock. Fan et al. [13] studied the thermal and neural behaviors of skin tissue subjected to thermal shock and introduced the concept of skin thermal shock resistance which is widely used for engineering materials. Shen et al. [14] examined the transfer of heat and elastic deformation in soft tissue using the Pennes bioheat transfer equation and modified Duhamel–Neuman equations. Afrin et al. [15] produced a model to describe heat transfer in living biological tissues, taking into account the properties of arterial and venous blood as well as tissue. It was concluded that the lag times are the same if the tissue and blood have the same properties, but different if their properties are different, and discussed how this model could be applied to brain and muscle tissues specifically.

Many researchers have applied the models of simple DPL to skin tissue to study its thermomechanical response. Youssef and Alghamdi [16] investigated how skin tissue responds to constant surface heat flux and found a mathematical solution using the two-temperature dual-phase-lag (TTDPL) model of bioheat transfer. Kumar et al. [17] used the DPL model to investigate the thermal behavior of living biological tissue with non-Fourier boundary conditions during thermal therapy where observed how different boundary conditions and coordinate systems affected surrounding healthy tissues and the temperature rise in peak hyperthermia when using certain types of boundary conditions with Cartesian coordinates. Xu et al. [18] studied the non-Fourier heat transfer process and its influence on the mechanical response in skin tissue using a DPL model to model bioheat transfer across the tissue and analyzed different surface heating boundary conditions with both exact solutions for single-layer models, as well as numerical solutions for multi-layer structural models. Liu et al. [19,20] analyzed an inverse non-Fourier bio-heat transfer problem in a bi-layer spherical geometry and compared their results with classical bio-heat transfer equations, then studied the bio-heat transfer problem in the skin using the DPL model to consider the effect of microstructural interaction.

Askarizadeh and Ahmadikia [21] utilized the DPL theory of biothermal transfer in handling the transient thermal transfer in skin tissue, whereas Kumar et al. [22] used a non-linear DPL biothermal transfer theory with a periodic thermal flow boundary condition to describe and simulate heat transport in skin tissues. Youssef and Alghamdi [23] introduced a mathematical analysis of thermoelastic skin tissue using the DPL heat conduction law and looked at three different types of heating. Li et al. [24] examined bioheat transport and thermal-produced mechanical behavior in bi-layered humanoid skin with varying heat material properties. Majchrzak and Stryczyński [25] analyzed the thermal and neural behaviors of skin tissue exposed to thermal loadings by introducing a new parameter called skin thermal shock resistance. Numerical simulations were performed which showed that skin resistance depends on Biot number, allowing it to be used in predicting both damage and pain caused by noxious heat treatments. Kumar et al. [26] display the heat damage that skin tissue sustained while being exposed to a moving thermal source and the DPL theory of biological heat transport is used to model the issue based on a memory-dependent derivative. Recently, Sobhy and Zenkour [27] presented a theory of skin tissue response that considered accounting for the effect of higher-order time derivatives based on the Lord and Shulman model.

The application of lasers for biothermal purposes has significantly expanded over the past half-century. In medical and clinical procedures, the application of lasers and heat transfers to the skin is essential. Therefore, controlling heat damage for the resulting use of lasers is crucial for treatment quality. The simple DPL model has been used in many laser applications. To estimate heat damage in organic tissue that has been exposed to laser radiation, Liu and Wang [28] employed the DPL model and assumed that the tissue would absorb and scatter laser light very significantly. Hobiny and Abbas [29] conducted a numerical study to illustrate the nonlinear behavior of the system and the effect of laser power intensity on its dynamic, and also estimated thermal damages to tissues based on the Arrhenius formulation. Zhang et al. [30] gave the generalized DPL model on skin tissue during laser irradiation and discussed the effects of some crucial parameters on the field variables. Alghamdi and Youssef [31] produced the bio-math model of the human eye under a decaying laser by applying the DPL theory. Moreover, it took into account the impacts of changes in blood perfusion, porosity, rate of evaporation, temperature ambient, and power density of laser irradiation. Zhou et al. [32] proposed the DPL model to improve the accuracy of predicting thermal damage during laser medical treatments. A semianalytical method was presented by Li et al. [33] to describe the problem of thermal transport in skin tissues exposed to time-varying laser heat and fluid cooling.

In the current article, the refined DPL theory is utilized to explain a ramp-type heat bio-thermal coupling issue that considered the effects of higher-order time derivatives. Ramp-type heat is a type of thermal loading that can be used to simulate the effects of exposure to a gradual increase/decrease in temperature over time on skin tissue. Ramp-type heating can be used in a variety of practical applications, such as medical treatments. For example, cryotherapy that during cryotherapy sessions affected area of the body is exposed to extremely low temperatures for a short period. This helps reduce inflammation and promote healing by decreasing blood flow and reducing nerve activity in the area being treated. Additionally, it may be used to treat skin conditions such as psoriasis and eczema by applying heat to the affected area. This helps reduce inflammation and improve healing time. The simple DPL, L–S, G–N II, and CTE theories are as well shown. The objective is to explain an issue of 1D and enhance a new generalized bio-thermal model. The effects of relaxation times, ramp-type heat parameters, the blood perfusion rate, and heat conductivity on different field variables during skin tissue are studied and characterized in figures within this study.

## 2. Basic Equations

Fourier’s law in connection with heat flux $\vec{q}$ to temperature gradient ∇θ, can be stated as
(1)$$\vec{q}\left(\vec{x},t\right)=-k_{t}\nabla \theta \left(\vec{x},t\right),$$
where $k_{t}$ denotes the heat conductivity parameter of tissue and $\theta =T-T_{b}$ represents the tissue temperature change, where $T_{b}$ denotes arterial blood temperature. Even though the traditional thermoelasticity (CTE) hypothesis has been much used because of its simple design and overall effective arrangement, the coupled theory built on Fourier’s conduction law has been amended to a justification for non-Fourier peculiarities in some media, such as living tissue and sand. Cattaneo [34] and Vernotte [35] revised the classical Fourier’s thermal conductivity law by proposing that includes relaxation time $\tau_{q}$ given by
(2)$$\vec{q}\left(\vec{x},t+\tau_{q}\right)=-k_{t}\nabla \theta \left(\vec{x},t\right),$$
where the relaxation time $\tau_{q}$ represents a phase lag (PL) of heat flux. The modification shows that for a specified position, the heat flux and temperature gradient happen at different moments. Additionally, known as Lord–Shulman (L–S) thermoelasticity theory. Tzou [6,36,37] has recommended the DPL model utilizing a more general type for the thermal conduction of Fourier’s, as illustrated below
(3)$$\vec{q}\left(\vec{x},t+\tau_{q}\right)=-k_{t}\nabla \theta \left(\vec{x},t+\tau_{\theta }\right),$$
where the relaxation time $\tau_{\theta }$ represents a PL of the temperature gradient and $(0\leq \tau_{\theta }<\tau_{q}).$ If we use Taylor’s series expansion of the above equation up to the higher-order terms in $\tau_{q}$ and $\tau_{\theta }$ [38,39,40,41,42,43,44,45,46], we obtain
(4)$$\left(1+\sum_{m=1}^{M}\frac{\tau_{q}^{m}}{m!}\frac{\partial^{m}}{\partial t^{m}}\right)\vec{q}=-k_{t}\left(1+\sum_{m=1}^{M}\frac{\tau_{\theta }^{m}}{m!}\frac{\partial^{m}}{\partial t^{m}}\right)\nabla \theta .$$

The energy balance equation is defined by [47,48,49]
(5)$$\rho_{t}c_{t}\frac{\partial \theta }{\partial t}+\gamma_{t}T_{b}\frac{\partial }{\partial t}\left(\mathrm{div}\vec{u}\right)=-\nabla \cdot \vec{q}+Q,$$
in which $\rho_{t}$ denotes the material density of the tissue, $c_{t}$ indicates the thermal capacity of a unit mass of the tissue, $\gamma_{t}=\left(2\mu_{t}+3\lambda_{t}\right)\alpha_{t}$ is the heat modulus where $\lambda_{t}$ and $\mu_{t}$ are Lamé’s constant of the tissue and $\alpha_{t}$ is the thermal expansion coefficient, $\vec{u}$ denotes displacement vector, $\mathrm{div}\vec{u}=e=e_{kk}$ represents volumetric strain, and $e_{ij}$ represents strain tensor. For biological tissue, the parameter Q is the heat source which has been widely adopted in many types of research such as [50,51], and is written as
(6)$$Q=w_{b}\rho_{b}c_{b}\left(T_{b}-T\right)+Q_{m}+Q_{L},$$
where $w_{b}$ is the blood perfusion rate, $\rho_{b}$ and $c_{b}$ indicate the specific heat and mass density of the blood. The terms $Q_{m}$ and $Q_{L}$ represent the thermal source of metabolic generation in tissue cells [52] and the external heat load [53], respectively. In arrangement with the energy conservative Equation (5) with Equation (4), one obtains the improved thermal conduction equation with two relaxation times as
(7)$$k_{t}\left(1+\sum_{m=1}^{M}\frac{\tau_{\theta }^{m}}{m!}\frac{\partial^{m}}{\partial t^{m}}\right)\nabla^{2}\theta \;=\left(\delta +\sum_{m=1}^{M}\frac{\tau_{q}^{m}}{m!}\frac{\partial^{m}}{\partial t^{m}}\right)\left[\frac{\partial }{\partial t}\left(\rho_{t}c_{t}\theta +\gamma_{t}T_{b}e\right)+w_{b}\rho_{b}c_{b}\theta -Q_{m}-Q_{L}\right],$$
which represents the refined DPL generalized thermoelasticity theory. The value of M is 3, which is conferring to the refined DPL theory required. The classical coupled thermoelasticity (CTE) theory is obtained by setting $\tau_{\theta }=\tau_{q}=0$ and δ=1 which the from
(8)$$k_{t}\nabla^{2}\theta =\left[\frac{\partial }{\partial t}\left(\rho_{t}c_{t}\theta +\gamma_{t}T_{b}e\right)+w_{b}\rho_{b}c_{b}\theta -Q_{m}-Q_{L}\right].$$

It should be noted that Biot [3] was the first to put forth the classical coupled thermoelasticity theory, which comprises both the aforesaid energy conservative equation and the upcoming equation of motion.

The simple DPL generalized thermoelasticity theory is achieved when M=1 and δ=1 in the form
(9)$$k_{t}\left(1+\tau_{\theta }\frac{\partial }{\partial t}\right)\nabla^{2}\theta =\left(1+\tau_{q}\frac{\partial }{\partial t}\right)\left[\frac{\partial }{\partial t}\left(\rho_{t}c_{t}\theta +\gamma_{t}T_{b}e\right)+w_{b}\rho_{b}c_{b}\theta -Q_{m}-Q_{L}\right].$$

However, the simple L–S theory is obtained from Equation (9) when $\tau_{\theta }=0$
(10)$$k_{t}\nabla^{2}\theta =\left(1+\tau_{q}\frac{\partial }{\partial t}\right)\left[\frac{\partial }{\partial t}\left(\rho_{t}c_{t}\theta +\gamma_{t}T_{b}e\right)+w_{b}\rho_{b}c_{b}\theta -Q_{m}-Q_{L}\right],$$
whereas the simple G–N II theory is obtained from Equation (7) when $\tau_{\theta }=\tau_{q}=0$, $\delta =\frac{\partial }{\partial t}$ and $k_{t}\to k_{t}^{*}$ where $k_{t}^{*}$ represents the rate of heat conductivity of the tissue as
(11)$$k_{t}^{*}\nabla^{2}\theta =\frac{\partial }{\partial t}\left[\frac{\partial }{\partial t}\left(\rho_{t}c_{t}\theta +\gamma_{t}T_{b}e\right)+w_{b}\rho_{b}c_{b}\theta -Q_{m}-Q_{L}\right],$$
which another generalized thermoelasticity model, developed by Green and Naghdi [54], can immediately provide for the non-Fourier impact of heat propagation without any relaxation time. Many articles [24,55], establish the value of $k_{t}^{*}$ as $k_{t}^{*}=c_{t}\left(\lambda_{t}+2\mu_{t}\right)/4$.

In the G–N II thermoelastic model, the rate of heat conductivity $k_{t}^{*}$ performs a key role in thermal wave propagation. Zhang et al. [56] found that the above value of $k_{t}^{*}$ is incompatible with organic material for its corresponding fast speed of the thermal wave. On the other hand, a small value of $k_{t}^{*}$ makes the G–N II model compatible with earlier thermoelastic models. Therefore, the value of the rate of heat conductivity could be shown as $k_{t}^{*}=0.04$ more suitably.

For the current 1D issue, the constitutive equations may be diminished to
(12)$$\sigma =\left(\lambda_{t}+2\mu_{t}\right)e-\gamma_{t}\theta ,$$
where
(13)$$e=\frac{\partial u}{\partial x},$$
and the equations of motion that do not take into account the body force are as follows
(14)$$\left(\lambda_{t}+2\mu_{t}\right)\frac{\partial^{2}u}{\partial x^{2}}-\gamma_{t}\frac{\partial \theta }{\partial x}=\rho_{t}\frac{\partial^{2}u}{\partial t^{2}}.$$

The above formula is used with Equation (8) to prepare the CTE theory, with Equation (10) to obtain the simple L–S theory, with Equation (11) to develop the simple G–N II theory, with Equation (9) to produce the simple DPL theory, and finally with Equation (7) to prepare the refined DPL theory.

## 3. Analytical Solution

Let us consider the modified DPL model, shown in Equations (7), (12) and (14) with neglecting $Q_{L}$, which may be expressed as
(15)$$\frac{\partial^{2}u}{\partial x^{2}}-a_{1}\frac{\partial \theta }{\partial x}=\frac{1}{C_{P}^{2}}\frac{\partial^{2}u}{\partial t^{2}},$$
(16)$$C_{T}^{2}\left(1+\sum_{m=1}^{M}\frac{\tau_{\theta }^{m}}{m!}\frac{\partial^{m}}{\partial t^{m}}\right)\frac{\partial^{2}\theta }{\partial x^{2}}=\left(\delta +\sum_{m=1}^{M}\frac{\tau_{q}^{m}}{m!}\frac{\partial^{m}}{\partial t^{m}}\right)\left[\left(w_{b}\rho_{c}+\frac{\partial }{\partial t}\right)\theta +\eta \frac{\partial^{2}u}{\partial t\partial x}\right]-Q_{0},$$
(17)$$\frac{\sigma }{\lambda_{t}+2\mu_{t}}=\frac{\partial u}{\partial x}-a_{1}\theta ,$$
where
(18)$$a_{1}=\frac{\gamma_{t}}{\lambda_{t}+2\mu_{t}},\;\;C_{P}^{2}=\frac{\lambda_{t}+2\mu_{t}}{\rho_{t}},\;\;C_{T}^{2}=\frac{k_{t}}{\rho_{t}c_{t}},\;\;\rho_{c}=\frac{\rho_{b}c_{b}}{\rho_{t}c_{t}},\;\;\eta =\frac{\gamma_{t}T_{b}}{\rho_{t}c_{t}},\;\;Q_{0}=\frac{\delta Q_{m}}{\rho_{t}c_{t}}.$$

Right now, we should introduce the initial and boundary conditions of the issue. The initial ones of the topic under consideration are believed to be
(19)$${\left.u\left(x,t\right)\right|}_{t=0}={\left.\frac{\partial^{n}u\left(x,t\right)}{\partial t^{n}}\right|}_{t=0}=0,\;\;\;\;{\left.\theta \left(x,t\right)\right|}_{t=0}={\left.\frac{\partial^{n}\theta \left(x,t\right)}{\partial t^{n}}\right|}_{t=0}=0,\;\;\;\;n\geq 1.$$

On both internal and external surfaces, the biological tissue is regarded as fixed. The heat load is applied on the external face of skin tissue while its internal face keeps the assessed temperature vanishing. Hence, the boundary conditions are expressed as
(20)$$\theta \left(0,t\right)=g\left(t\right),\;\;\;\;\theta \left(L,t\right)=0,\;\;\;\;\sigma \left(0,t\right)=0,\;\;\;\;\sigma \left(L,t\right)=0,$$
where $g\left(t\right)$ is the thermal loading on the higher face of the skin tissue x=0, as depicted in Figure 1. Second, one assumes that the plane x=0 of the tissue is under ramp-type heat as
(21)$$g\left(t\right)=\theta_{0}\left\{\begin{array}{lll} \frac{t}{t_{0}} & \mathrm{if} & 0<t<t_{0}\\ 1 & \mathrm{if} & \;\;\;\;\;\;\;t\geq t_{0} \end{array}\right.$$
where $\theta_{0}>0$ is the constant that signifies the thermal loading and $t_{0}>0$ is the ramp-type heat parameter.

## 4. Laplace Transforms

Taking the Laplace transform described in the context of
(22)$$\bar{H}\left(x,s\right)=L\left\{H\left(x,t\right)\right\}=\int_{0}^{\infty }H\left(x,t\right)e^{-st}dt,$$
to the two sides of Equations (15)–(17) and employing initial conditions (19), one obtains field equations in Laplace change space as
(23)$$\left(\frac{d^{2}}{dx^{2}}-2a_{2}\right)\bar{u}-a_{1}\frac{d\bar{\theta }}{dx}=0,$$
(24)$$\left(\frac{d^{2}}{dx^{2}}-2a_{3}\right)\bar{\theta }=2a_{4}\frac{d\bar{u}}{dx}-{\bar{Q}}_{1},$$
(25)$$\frac{\bar{\sigma }}{\lambda_{t}+2\mu_{t}}=\frac{d\bar{u}}{dx}-a_{1}\bar{\theta },$$
where
(26)$$a_{2}=\frac{s^{2}}{2C_{P}^{2}},\;\;\;\;a_{3}=\frac{\left(w_{b}\rho_{c}+s\right)\left(\bar{\delta }+\sum_{m=1}^{M}\frac{\tau_{q}^{m}}{m!}s^{m}\right)}{2C_{T}^{2}\;\left(1+\sum_{m=1}^{M}\frac{\tau_{\theta }^{m}}{m!}s^{m}\right)},\;a_{4}=\frac{\eta s\left(\bar{\delta }+\sum_{m=1}^{M}\frac{\tau_{q}^{m}}{m!}s^{m}\right)}{2C_{T}^{2}\;\left(1+\sum_{m=1}^{M}\frac{\tau_{\theta }^{m}}{m!}s^{m}\right)},\;\;\;\;{\bar{Q}}_{1}=\frac{{\bar{Q}}_{0}}{sC_{T}^{2}\;\left(1+\sum_{m=1}^{M}\frac{\tau_{\theta }^{m}}{m!}s^{m}\right)}.$$

It is stated that the over bar icon denoted its Laplace transform and s suggests the Laplace parameter. Equations (23) and (24) are explained in the Laplace domain to attain
(27)$$\bar{\theta }=\sum_{i=1}^{2}\left(A_{i}\;e^{\xi_{i}x}+B_{i}\;e^{-\xi_{i}x}\right)+{\bar{Q}}_{2},$$
(28)$$\bar{u}=\sum_{i=1}^{2}\beta_{i}\left(A_{i}\;e^{\xi_{i}x}-B_{i}\;e^{-\xi_{i}x}\right),$$
where $A_{i}$ and $B_{i}$ are constants depending on s and ${\bar{Q}}_{2}={\bar{Q}}_{1}/2a_{3}$. The parameters $\xi_{i}$ and $\beta_{i}$ are represented as
(29)$$\xi_{1},\xi_{2}=\sqrt{a_{1}a_{4}+a_{2}+a_{3}\pm \xi_{0}},\;\xi_{0}=\sqrt{{\left(a_{1}a_{4}+a_{2}\right)}^{2}+a_{3}\left[a_{3}+2\left(a_{1}a_{4}-a_{2}\right)\right]},$$
(30)$$\beta_{i}=\frac{\xi_{i}\left(\xi_{i}^{2}-2a_{1}a_{4}-2a_{3}\right)}{4a_{2}a_{4}}.$$

Moreover, the dilatation in Equation (13) is provided in the Laplace domain by
(31)$$\bar{e}=\sum_{i=1}^{2}\beta_{i}\xi_{i}\left(A_{i}\;e^{\xi_{i}x}+B_{i}\;e^{-\xi_{i}x}\right).$$

Equation (25) also produces the following results for the axial stress
(32)$$\bar{\sigma }=\sum_{i=1}^{2}\zeta_{i}\left(A_{i}\;e^{\xi_{i}x}+B_{i}\;e^{-\xi_{i}x}\right)-{\bar{Q}}_{3},$$
where
(33)$$\zeta_{i}=\left(\lambda_{t}+2\mu_{t}\right)\left(\beta_{i}\xi_{i}-a_{1}\right),\;\;\;\;{\bar{Q}}_{3}=\left(\lambda_{t}+2\mu_{t}\right)a_{1}{\bar{Q}}_{2}.$$

In the Laplace transform domain, the conditions that appeared in Equation (20) are expressed as
(34)$${\left.\bar{\theta }\left(x,s\right)\right|}_{x=0}=\frac{\theta_{0}\left(1-e^{-t_{0}s}\right)}{t_{0}s^{2}}={\bar{G}}_{s},$$
(35)$${\left.\bar{\theta }\left(x,s\right)\right|}_{x=L}=0,\;\;\;\;{\left.\bar{\sigma }\left(x,s\right)\right|}_{x=0,L}=0.$$

The solution of the prodigious composition of immediate conditions offers the unknown parameters $A_{i}$ and $B_{i}$. Employing the above conditions on Equations (27) and (28), one achieves
(36)$$\left[\begin{array}{cc} \begin{array}{cc} 1 & 1\\ e^{\xi_{1}L} & e^{-\xi_{1}L} \end{array} & \begin{array}{cc} 1 & 1\\ e^{\xi_{2}L} & e^{-\xi_{2}L} \end{array}\\ \begin{array}{cc} \xi_{1} & \xi_{1}\\ \xi_{1}e^{\xi_{1}L} & \xi_{1}e^{-\xi_{1}L} \end{array} & \begin{array}{cc} \xi_{2} & \xi_{2}\\ \xi_{2}e^{\xi_{2}L} & \xi_{2}e^{-\xi_{2}L} \end{array} \end{array}\right]\left\{\begin{array}{c} \begin{array}{c} A_{1}\\ B_{1} \end{array}\\ \begin{array}{c} A_{2}\\ B_{2} \end{array} \end{array}\right\}=\left\{\begin{array}{c} \begin{array}{c} {\bar{G}}_{s}-{\bar{Q}}_{2}\\ -{\bar{Q}}_{2} \end{array}\\ \begin{array}{c} {\bar{Q}}_{3}\\ {\bar{Q}}_{3} \end{array} \end{array}\right\}.$$

To complete the solutions in the Laplace transform domain, one solves the above system of linear equations to obtain
(37)$$A_{1}=\frac{\left(\zeta_{2}{\bar{Q}}_{2}+{\bar{Q}}_{3}\right)e^{\xi_{1}L}+\zeta_{2}\left({\bar{G}}_{s}-{\bar{Q}}_{2}\right)-{\bar{Q}}_{3}}{\left(\zeta_{1}-\zeta_{2}\right)\left(e^{2\xi_{1}L}-1\right)},\;\;\;\;B_{1}=-\frac{\left[\left(\zeta_{2}\left({\bar{G}}_{s}-{\bar{Q}}_{2}\right)-{\bar{Q}}_{3}\right)e^{\xi_{1}L}+\zeta_{2}{\bar{Q}}_{2}+{\bar{Q}}_{3}\right]e^{\xi_{1}L}}{\left(\zeta_{1}-\zeta_{2}\right)\left(e^{2\xi_{1}L}-1\right)},\;A_{2}=\frac{\zeta_{1}\left({\bar{Q}}_{2}-{\bar{G}}_{s}\right)-\left(\zeta_{1}{\bar{Q}}_{2}+{\bar{Q}}_{3}\right)e^{\xi_{2}L}+{\bar{Q}}_{3}}{\left(\zeta_{1}-\zeta_{2}\right)\left(e^{2\xi_{2}L}-1\right)},\;\;\;\;B_{2}=\frac{\left[\left(\zeta_{1}\left({\bar{G}}_{s}-{\bar{Q}}_{2}\right)-{\bar{Q}}_{3}\right)e^{\xi_{2}L}+\zeta_{1}{\bar{Q}}_{2}+{\bar{Q}}_{3}\right]e^{\xi_{2}L}}{\left(\zeta_{1}-\zeta_{2}\right)\left(e^{2\xi_{2}L}-1\right)}.$$

This completes the solution to the issue in the transform domain. Since the statements in Equations (27) and (28) are extremely difficult, it is quite hard to obtain the inverse transform methodically in the time domain. Hence, the Laplace transform inversion method will be utilized to achieve the behaviors of field variables in the definite time domain. To accomplish numerical outcomes in the physical domain, we utilize the Riemann-sum approximation method. In this approach, any function $\bar{H}\left(x,z,s\right)$ in Laplace domain is upset to physical domain $H\left(x,z,t\right)$ by applying the notable equation [57]
(38)Hx,z,t=eϱtt12ReH¯x,z,ϱ+Re∑n=0NH¯x,z,ϱ+nπIt(−1)n,
in which Re is the real part of a function and $I=\sqrt{-1}$ is the imaginary number unit. Various numerical experiments have confirmed that the value ϱ fulfills the relation ϱt≈4.7 for rapid convergence [36].

## 5. Numerical Results and Discussion

Here, is a discussion of the numerical outcomes for all variables through skin tissue. The modification was compared with different thermoelastic theories in predicting the thermal responses of tissues. The impact of ramp-type heat coefficient, phase-lags, blood perfusion rate, and heat conductivity are examined in the expectation of thermal and elastic responses. Table 1 provides a list of the thermophysical features of blood and biological tissue [58,59,60]. The thickness of the skin tissue L=1 mm and $\theta_{0}=1$. The values of temperature $\left(\theta \right)$, displacement $\left(u\right)$, volumetric strain $\left(e\right)$, and axial stress $\left(\sigma \right)$ are described concerning Equation (38). The numerical outcomes are taken and discussed in detail in Figure 2, Figure 3, Figure 4, Figure 5, Figure 6, Figure 7, Figure 8, Figure 9, Figure 10, Figure 11, Figure 12, Figure 13, Figure 14 and Figure 15.

### 5.1. Refined DPL, Simple DPL, L–S, G–N II, and Classical Models

Figure 2, Figure 3, Figure 4 and Figure 5 show the field quantities resulting from various coupling theorems with fixed relaxation times $\tau_{\theta }=1.8$, $\tau_{q}=2.8$, and time t=3.

Figure 2 considers different theories to display the temperature $\left(\theta \right)$ distributions through the thickness of the skin tissue. The temperature reduces directly with the increase in x for the CTE, simple DPL, and modified DPL generalized thermoelastic theories. The response of the L–S, and G–N II thermoelastic theories has a different look. The modified DPL generalized thermoelastic theory falls between the L–S and the simple DPL. Another obvious in thermal response is the temperature is higher in the affected zone by the CTE model than in other theories.

Figure 3 shows the displacement distributions across the thickness of the skin tissue in the context of different theories. Before x=0.47 the displacement increases directly as x increases, whereas after x=0.47, we obtain more displacement along x direction due to generalized thermal expansion for CTE, simple DPL, and refined DPL. In the cases of G–N II model before x=0.29 and L–S model before x=0.39 the displacement increases directly as x increases, whereas after these positions the displacement increases along x direction.

It is exciting to see that the displacement $\left(u\right)$ has the same value when x=0.47 for CTE, simple DPL, and refined DPL generalized. As the temperature is higher in the affected zone in the CTE model Figure 2, consequently, the displacement in the CTE model is larger than in other theories.

Figure 4 shows the dilatation distributions $\left(e\right)$ across the skin tissue by applying various theories. The dilatation reduces directly with the increase in x for CTE, simple DPL, and refined DPL theory, whereas the L–S, and G–N II theories have a different look. The behavior of $\left(e\right)$ using the refined DPL theory falls between the L–S and the simple DPL. The dilatation is higher in the affected zone by the CTE model than in other theories.

Figure 5 displays the stress distributions across the thickness of the skin tissue according to different theories. The stress is no longer decreasing directly as increases and has its minimum value at x=0.49 for CTE, simple DPL generalized thermoelastic models, whereas the minimum value for refined DPL occurs at x=0.46. The performance due to the G–N II and L–S theories looks different. The L–S generalized thermoelastic model gives the lowest stresses.

### 5.2. Effect of Ramp-Type Heat

Considering the impacts of the ramp-type heat parameter on the distributions of the temperature, displacement, dilatation, and stress for refined DPL generalized thermoelastic model for numerous values of the ramp-type heating parameter $t_{0}=2.3$, 2.8, 3.3, when time t=2.8 and $\theta_{0}=1$ in Figure 6, Figure 7, Figure 8 and Figure 9.

The ramp-type heat parameter has a significant effect on the distributions of temperature, displacement, dilatation, and stress. Figure 6 shows that the two cases $t>t_{0}$ and $t=t_{0}$ the temperature equal to one when x=0, whereas the temperature when $t<t_{0}$ equal 0.84. For all cases, the values of the temperature go to zero at x=L, which agrees with the mechanical boundary conditions on both surfaces. In general, the temperature reduces with the increase in the values of the ramp-type heating parameter.

Figure 7 shows the variations of displacement across the thickness of the skin tissue using numerous values of the ramp-time parameter for the refined DPL model, the displacement vanishes when x=0.12. However, before this position, the displacement increases as $t_{0}$ increases and vice versa after this position.

Figure 8 shows the variations of dilatation $\left(e\right)$ across the thickness of the skin tissue utilizing a variety of values of the ramp-time parameter for the refined DPL model. For cases $t>t_{0}$ and $t=t_{0}$ the dilatation equal to 0.28 when x=0, whereas the dilatation at case $t<t_{0}$ equal 0.23. In general, the dilatation e reduces with the increase in the values of the ramp-type heating parameter.

Figure 9 shows the variations in stress $\left(\sigma \right)$ across the thickness of the skin tissue utilizing a variety of values of ramp-time heat parameters for the refined DPL model. The stress vanishes at the skin tissue edges which agrees with the mechanical boundary conditions on both surfaces. For cases $t>t_{0}$ and $t=t_{0}$, we have the same shape of the stress curve, noting that the lowest stress value is obtained at $t>t_{0}$, whereas the stress distribution is different at case $t<t_{0}$. It is exciting to see that the stress has the same value when x=0.45 for cases $t=t_{0}$ and $t<t_{0}$.

### 5.3. Impact of the Thermal Relaxation Times

In DPL thermoelasticity the relaxation times are important parameters in establishing the thermal behaviors in organic tissue. The graphs in Figure 10, Figure 11 and Figure 12 depict the variations of temperature, displacement, and stress under numerous phase lags when the ramp-type heating parameter remains constant $t_{0}=2.5$ with time t=2. Figure 10a, Figure 11a and Figure 12a showed $\tau_{q}=1.7$ with different levels of $\tau_{\theta },$ whereas Figure 10b, Figure 11b and Figure 12b showed $\tau_{\theta }=1.2$ with different levels of $\tau_{q}$.

Figure 10 shows the changes in temperature $\left(\theta \right)$ due to the refined DPL model under (a) different $\tau_{\theta }$ with fixed $\tau_{q}=1.7$ and (b) different $\tau_{q}$ with fixed $\tau_{\theta }=1.2$. It is clear that the temperature is sensitive to the change of both phase lags. In Figure 10a, the temperature is shown to increase with the increase in $\tau_{\theta }$. Additionally, in Figure 10b, the temperature decreases with the increase in $\tau_{q}$. This implies that the two-phase lags have opposite effects on the temperature.

Figure 11 shows the distributions of displacement $\left(u\right)$ due to the refined DPL model under (a) different $\tau_{\theta }$ with fixed $\tau_{q}=1.7$ and (b) different $\tau_{q}$ with fixed $\tau_{\theta }=1.2$. Once again, displacement u is sensitive to the variation of both phase lags, the displacement vanishes when x=0.44. However, before this position, the displacement u decreases as $\tau_{\theta }$ increases and vice versa after this position (Figure 11a). The exact opposite happens in (Figure 11b). This implies that the two-phase lags have opposite effects on the displacement.

Figure 12 shows the distributions of stress $\left(\sigma \right)$ due to the refined DPL model under (a) different $\tau_{\theta }$ with fixed $\tau_{q}=1.7$ and (b) different $\tau_{q}$ with fixed $\tau_{\theta }=1.2$. It is interesting to see that stress is sensitive to the variation of both phase lags. The stress σ decreases as $\tau_{\theta }$ increases (Figure 12a) and as $\tau_{q}$ decreases (Figure 12b). This implies that the two-phase lags have opposite effects on stress.

### 5.4. Impact of Blood Perfusion Rate

The impact of blood perfusion, which represents the thermal energy exchange between tissue and generally flowing blood, is a further difference in thermoelastic behavior in organic tissue. Figure 13 shows the distributions of (a) temperature θ and (b) displacement u across the skin tissue using various values of blood perfusion rate due to the refined DPL model.

Some portions are enlarged in Figure 13. It can be seen from Figure 13a that the temperature θ increases as blood perfusion $w_{b}$ decreases. However, it can be found from Figure 13b that before x=0.42, the displacement u increases as blood perfusion $w_{b}$ increases, whereas the opposite happens afterward.

### 5.5. Impact of Heat Conductivity

The impact of heat conductivity, which represents the measure of how well heat energy can be transferred through a tissue, is also a further difference in thermoelastic behavior in organic tissue. Figure 14 shows the distributions of (a) temperature θ and (b) displacement u across the skin tissue using various values of heat conductivity $k_{t}$ due to the refined DPL model.

It is clear from Figure 14a that larger values of $k_{t}$ results in greater temperature, which results in a smoother temperature distribution curve, meaning that temperature changes occur gradually and consistently instead of drastically. Figure 14b, the displacement vanishes when x=0.48. However, before this position, the displacement u decreases as $k_{t}$ increases and vice versa after this position.

For the sake of completion, extra Figure 15 and Figure 16 are provided to illustrate the 3D distributions of temperature $\theta \left(x,t\right)$ against time and through the thickness of the skin tissue. Five theories are used for this purpose. It is noticed that CTE and simple DPL theory exhibit similar behavior, although the behavior of the L–S, and G–N II is almost similar, the distribution of temperature is different for the refined DPL theory. As we have seen in 2D, the refined DPL theory gives a distribution in the middle where L–S and G–N II are located under it, whereas CTE and simple DPL are above it.

## 6. Conclusions

In the present work, we believe the newly utilized model of thermoelasticity based on a DPL to be revealed in the thermal conduction equation through skin tissue subjected to a ramp-type heat. The refined DPL thermoelasticity model is a higher-order time-derivative theory that considers accounting for the effect of multiple time derivatives which differs from generalized thermoelasticity theories. One-dimensional skin tissue was considered with a small thickness and its outer surface traction free, whereas its inner surface had no temperature increment or traction. The proposed model’s system of governing equations is created using generalized thermoelasticity theory. Laplace transform techniques were used to analyze these data, which were then inverted using the Tzuo method to calculate numerical results in the time domain. The former generalized thermoelastic theories are obtained from the present model. Comparisons between the refined DPL generalized thermoelastic model and existent generalized thermoelastic theories have been performed. The analytical and numerical analyses of governing equations both show a significant influence on the ramp-type heating coefficient, relaxation times, blood perfusion rate, heat conductivity, and time parameters. We have noticed that the values of the studied parameters decrease when the values of the ramp-type heat parameter increase, whereas the phase lags have opposite to each other effects. Some portions are enlarged to show the effect of the blood perfusion rate $w_{b}$. The heat conductivity $k_{t}$ significantly affects the temperature distribution and a lower value is more advantageous in some applications, where a slower rate of temperature change may be due to a desire to avoid tissue damage. In particular, increases in the $k_{t}$ lead to the temperature change happening more quickly, and a larger displacement being created due to the heat. Last but not least, the proposed model might be used for many applications in bioheat transfer applications.

## Figures and Tables

**Figure 1 materials-16-02421-f001:**
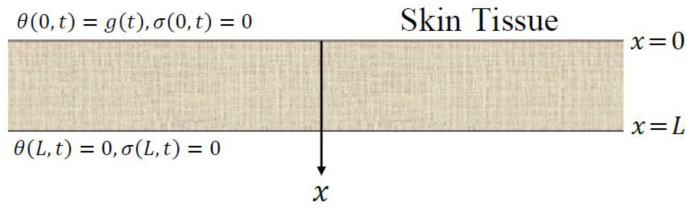
The skin tissue pattern with boundary conditions.

**Figure 2 materials-16-02421-f002:**
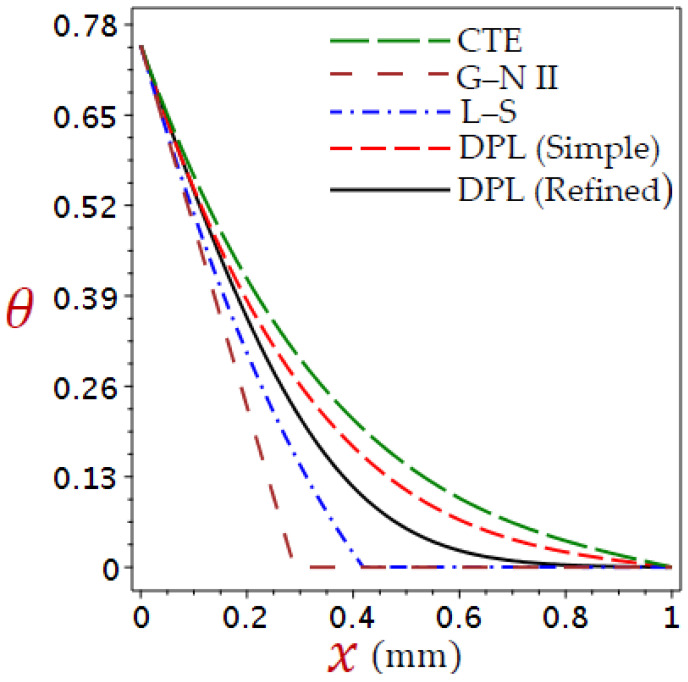
Comparison of variation of temperature $\left(\theta \right)$ across the skin tissue utilizing numerous theories.

**Figure 3 materials-16-02421-f003:**
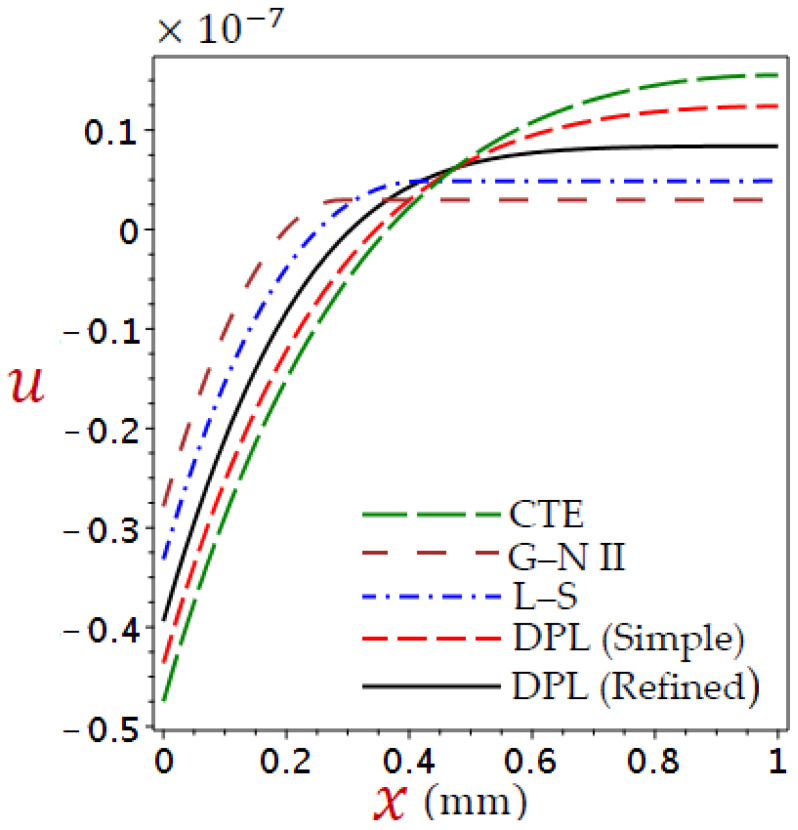
Comparison of variation of displacement $\left(u\right)$ across the skin tissue utilizing numerous theories.

**Figure 4 materials-16-02421-f004:**
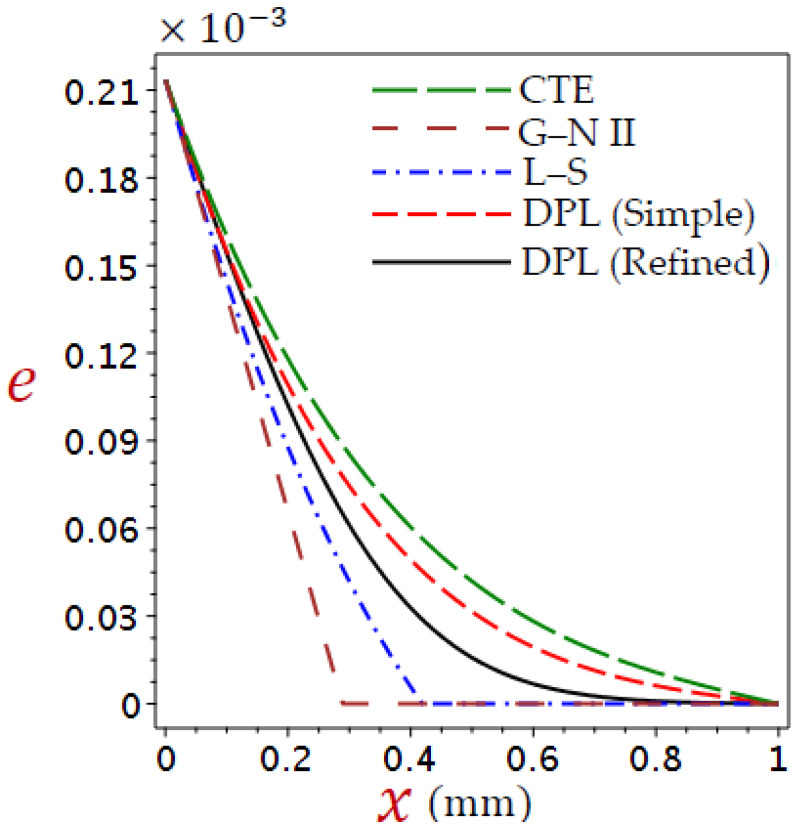
Comparison of variation of dilatation $\left(e\right)$ across the skin tissue utilizing numerous theories.

**Figure 5 materials-16-02421-f005:**
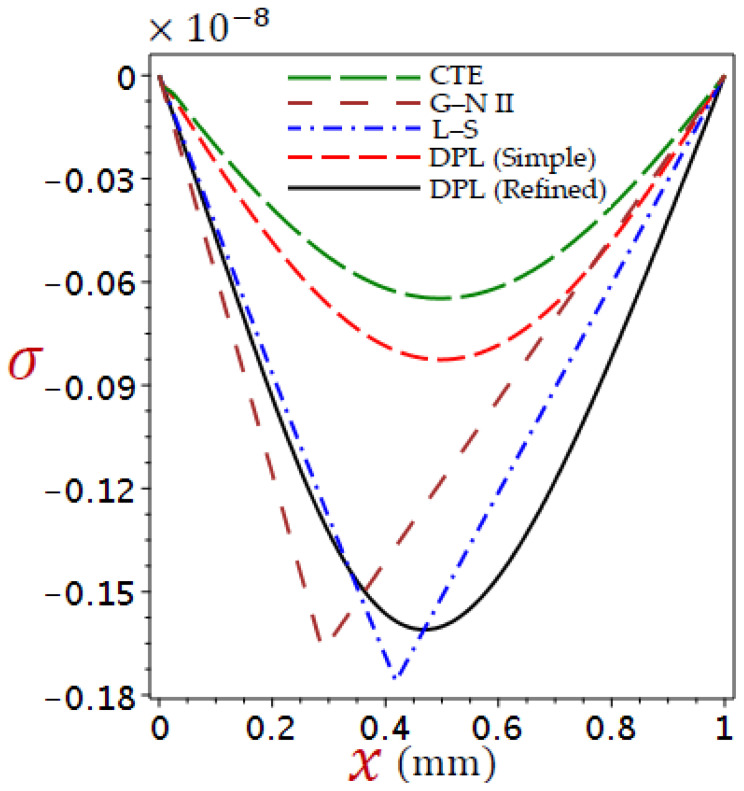
Comparison of variation of stress $\left(\sigma \right)$ across the skin tissue utilizing numerous theories.

**Figure 6 materials-16-02421-f006:**
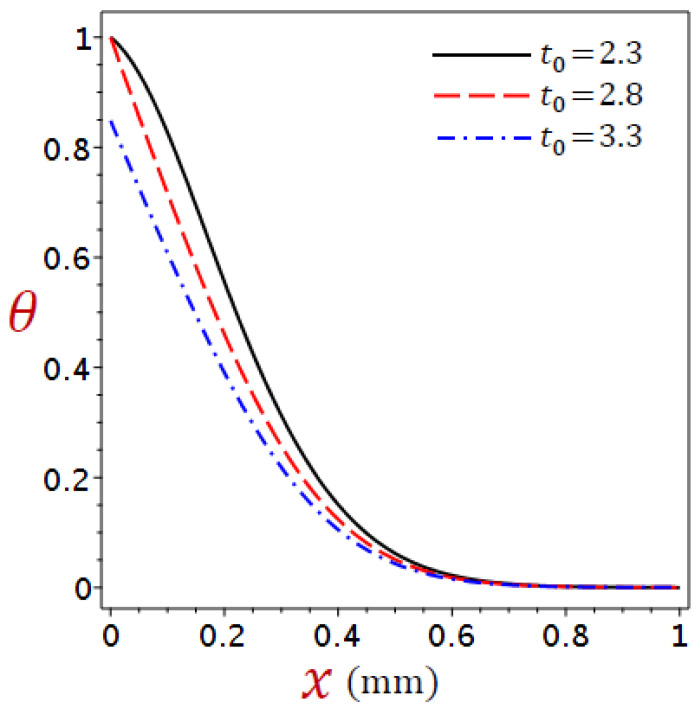
Variation of temperature $\left(\theta \right)$ across the skin tissue using numerous values of the ramp-time parameter for refined DPL model.

**Figure 7 materials-16-02421-f007:**
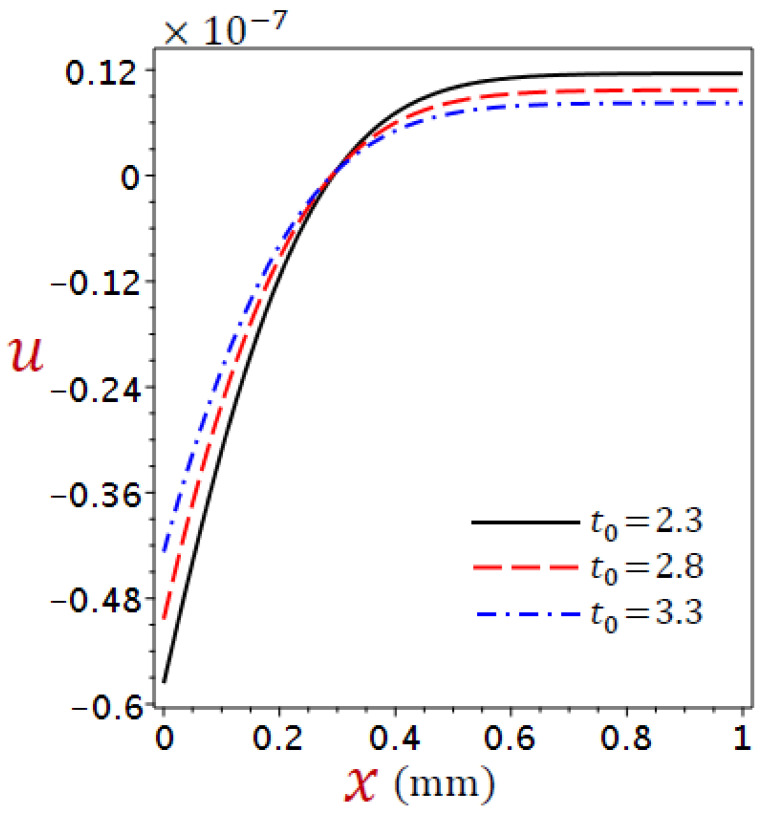
Variation of displacement $\left(u\right)$ across the skin tissue using numerous values of the ramp-time parameter for the refined DPL model.

**Figure 8 materials-16-02421-f008:**
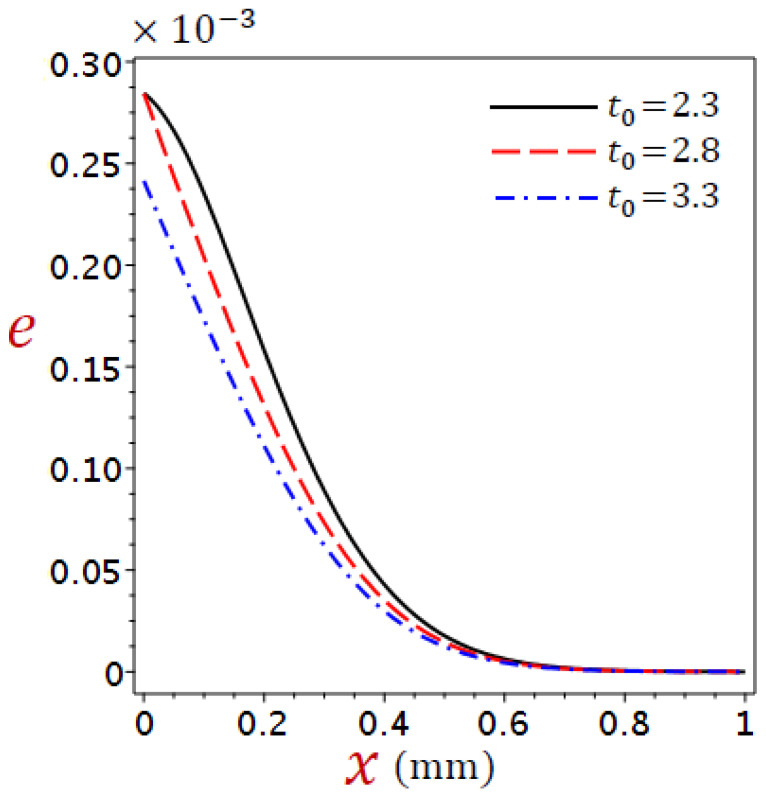
Variations of dilatation $\left(e\right)$ across the skin tissue using numerous values of the ramp-time parameter for the refined DPL model.

**Figure 9 materials-16-02421-f009:**
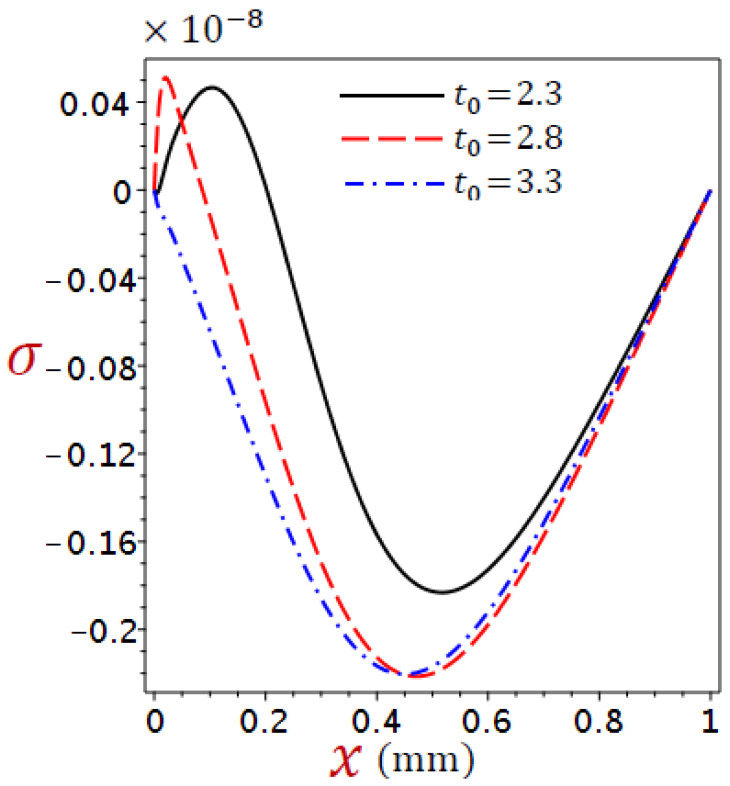
Variations of stress $\left(\sigma \right)$ across the skin tissue using numerous values of the ramp-time parameter for the refined DPL model.

**Figure 10 materials-16-02421-f010:**
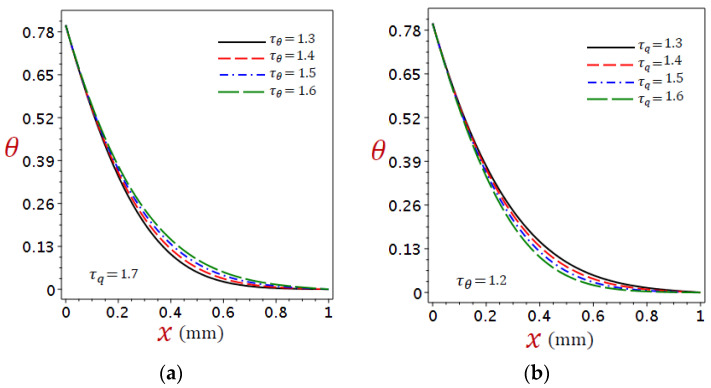
Distributions of temperature $\left(\theta \right)$ for refined DPL model under (**a**) different $\tau_{\theta }$ with $\tau_{q}=1.7$ and (**b**) different $\tau_{q}$ with $\tau_{\theta }=1.2$.

**Figure 11 materials-16-02421-f011:**
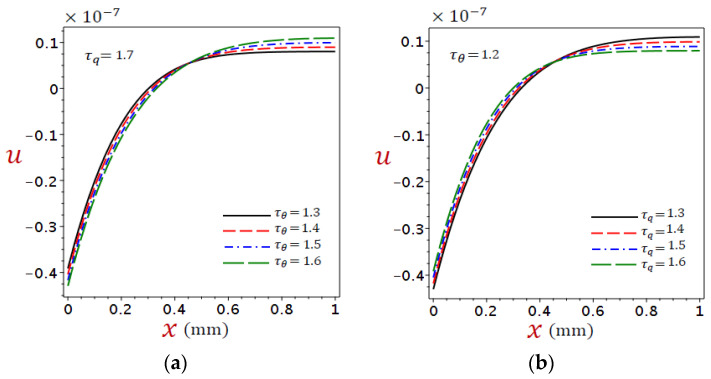
Distributions of displacement $\left(u\right)$ for refined DPL model under (**a**) different $\tau_{\theta }$ with $\tau_{q}=1.7$ and (**b**) different $\tau_{q}$ with $\tau_{\theta }=1.2$.

**Figure 12 materials-16-02421-f012:**
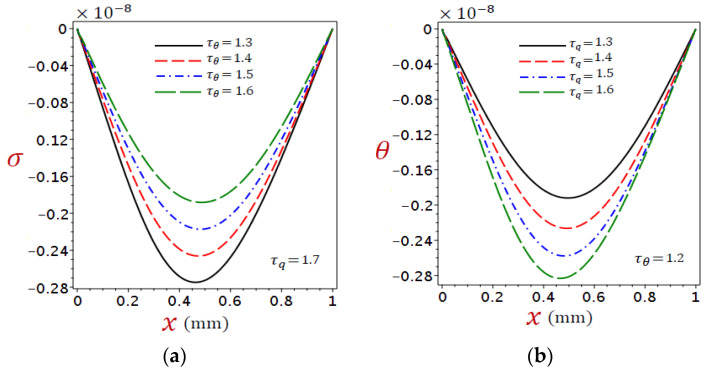
Distributions of stress $\left(\sigma \right)$ for refined DPL model under (**a**) different $\tau_{\theta }$ with $\tau_{q}=1.7$ and (**b**) different $\tau_{q}$ with $\tau_{\theta }=1.2$.

**Figure 13 materials-16-02421-f013:**
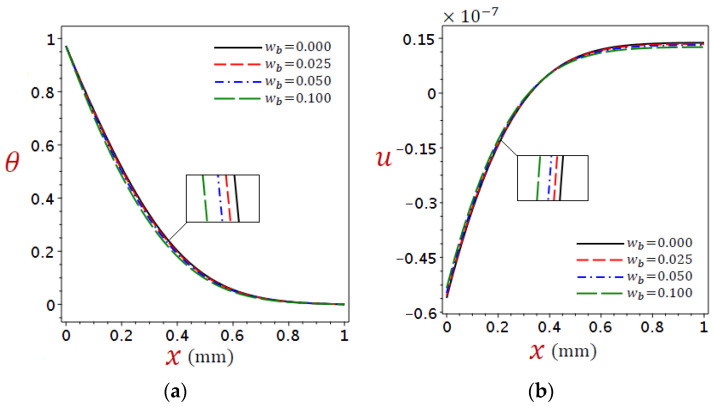
Distributions of (**a**) temperature $\left(\theta \right)$ and (**b**) displacement $\left(u\right)$ across the skin tissue using numerous values of blood perfusion rate $\left(w_{b}\right)$ due to the refined DPL model.

**Figure 14 materials-16-02421-f014:**
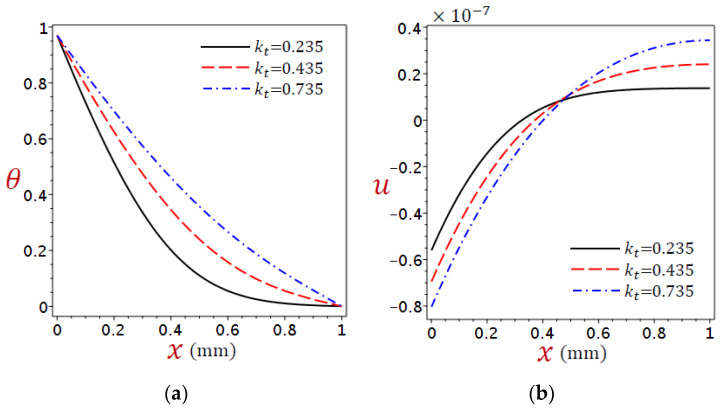
Distributions of (**a**) temperature $\left(\theta \right)$ and (**b**) displacement $\left(u\right)$ across the skin tissue using numerous values of heat conductivity $\left(k_{t}\right)$ due to the refined DPL model.

**Figure 15 materials-16-02421-f015:**
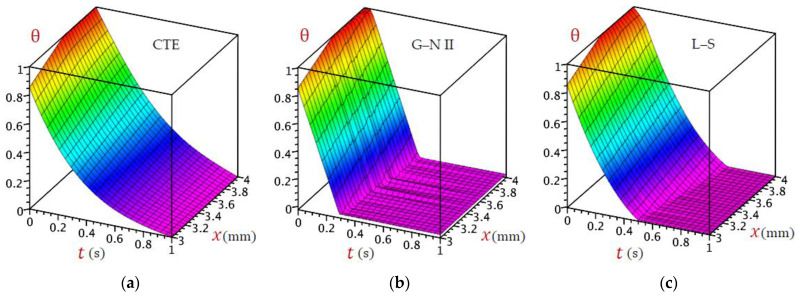
Three-dimensional distributions of temperature $\left(\theta \right)$ for (**a**) CTE, (**b**) G–N II, and (**c**) L–S models.

**Figure 16 materials-16-02421-f016:**
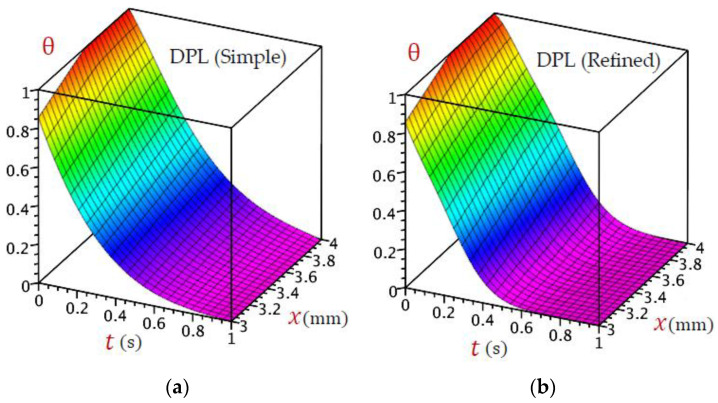
Three-dimensional distributions of temperature $\left(\theta \right)$ for (**a**) simple DPL and (**b**) refined DPL models.

**Table 1 materials-16-02421-t001:** Thermophysical features of blood and biological tissue.

Feature	Value	Unit
Lame’s constant $\lambda_{t}$	$8.27\times 10^{8}$	$\mathrm{kg}/\left({m\;s}^{2}\right)\;$
Lame’s constant $\mu_{t}$	$3.446\times 10^{7}$	$\mathrm{kg}/\left({m\;s}^{2}\right)$
Density $\rho_{t}$	1190	$\mathrm{kg}/m^{3}$
Tissue’s specific heat $c_{t}$	3600	$J/\left(K\;\mathrm{kg}\right)$
Thermal conductivity $k_{t}$	0.235	$W/\left(m\;K\right)$
Blood density $\rho_{b}$	1060	$\mathrm{kg}/m^{3}$
Blood’s specific heat $c_{b}$	3770	$J/\left(K\;\mathrm{kg}\right)$
Linear thermal expansion $\alpha_{t}$	$1\times 10^{-4}$	1/K
Arterial blood temperature $T_{b}$	310	K
Rate of blood perfusion $w_{b}$	$1.87\times 10^{3}$	1/s
Metabolic heat source $Q_{m}$	368.1	$W/m^{3}$

## Data Availability

Not applicable.

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
