# Peer review of "Refined Dual-Phase-Lag Theory for the 1D Behavior of Skin Tissue under Ramp-Type Heating"

_materials, 2023, doi:10.3390/ma16062421_

Round 1
Reviewer 1 Report
The paper should precisely indicate the difference and improvement of the existing DPL model. Please also describe the research tool (application). Conclusions to the work are too general, please specify them and emphasize the authors' achievements.
Reviewer 2 Report
Journal: Materials (ISSN 1996-1944)
Manuscript ID: materials-2250846
Title: Refined Dual-Phase-Lag Theory for the 1D Behavior of Skin Tissue Under Ramp-Type Heating
Reviews:
In this paper, a coupled thermoelasticity analysis in a thin skin tissue has been carried out using an analytical approach based on a refined dual-phase-lag (DPL) thermal conduction theory. There are some comments on the presented analysis in this article:
- The employed refined dual-phase-lag (DPL) thermal conduction theory was proposed for the solids not for the blood and biological tissue. How to prove that the refined dual-phase-lag (DPL) thermal conduction theory is proper for the skin tissue? In other words, a proper themoelasticity model should be developed for the material like skins.
- Is there any experimental evidence on the thermo-mechanical behaviors of the skin?
- What is the reference of thermophysical features of blood and biological tissue?
- I suggest to add some comparisons (preferably with experimental results) to verify the reported results and data.
- What is the practical example of the ramp-type heating on the skin?
- I recommend to extend the introduction section on the application of various coupled thermoelasticity theories by citing some references
Reviewer 3 Report
The scientific paper "Refined Dual-Phase-Lag Theory for the 1D Behavior of Skin Tissue Under Ramp-Type Heating” aimed to demonstrate a mathematical analysis of thermoelastic skin tissue presented based on a refined theory of dual-phase-lag (DPL) thermal conduction. It can be considered that:
1) The abstract must be rewritten. The results in summary form should be presented, as well as a better description of the methodology.
2) In all manuscripts, when the authors of an article are cited, add the year of publication. An example: replace in line 41 "Singh and Melnik [2] have reviewed...." with "Singh and Melnik (2020) [2] have reviewed....."
3) The introduction is fragmented into several very brief quotes from previous studies that supported the research. In lines 49-65, the names of the authors are cited seven times. The form of writing is poor and should be improved.
4) The number of figures is excessive (19). I suggest joining some images on a single plate.
5) The reproducibility of the experiment is not clear, as well as its clinical application.
Round 2
Reviewer 3 Report
The changes made were not sufficient to improve the quality of the manuscript.
Author Response
Firstly, we would like to thank Reviewer 3 for his efforts and time spent reviewing this paper.
Once again the hole manuscript is modified:
- The Intro is re-formatted and augmented.
- Some explanations are added.
- The MSC are added.
- All figures are checked.
- The Reference list is re-formatted and rearranged according the Intro's modifications.